# Enhancing the Properties of FeSiBCr Amorphous Soft Magnetic Composites by Annealing Treatments

**Hongya Yu** [1,2,3,*], **Jiaming Li** [2], **Jingzhou Li** [1], **Xi Chen** [1,2,3], **Guangze Han** [1,2,3], **Jianmin Yang** [1] and **Rongyin Chen** [1]

[1] Dongguan Mentech Optical & Magnetic Co., Ltd., Dongguan 523330, China; justin.li@mnc-tek.com.cn (J.L.); xichen@scut.edu.cn (X.C.); phgzhan@scut.edu.cn (G.H.); jimi.yang@mnc-tek.com.cn (J.Y.); chen.rong.yin@mnc-tek.com.cn (R.C.)

[2] School of Materials Science and Engineering, South China University of Technology, Guangzhou 510640, China; johnlee1215@163.com

[3] South China Institute of Collaborative Innovation, Dongguan 523808, China

\* Correspondence: yuhongya@scut.edu.cn

**Abstract:** Fe-based amorphous powder cores (AMPCs) were prepared from FeSiBCr amorphous powders with phosphate–resin hybrid coating. The high-frequency magnetic properties of AMPCs annealed at different temperatures were systematically studied. After annealing at low temperatures, the effective permeability and core loss improved due to the internal stress of the powder cores being released. The sample annealed at 480 °C exhibits the lowest hysteresis loss of about 29.6 mW/cm$^3$ at 800 kHz as well as a maximum effective permeability of 36.4, remaining stable until 3 MHz, which could be useful for high-frequency applications.

**Keywords:** amorphous powder cores; high frequency; internal stress; annealing

## 1. Introduction

Soft magnetic composites (SMCs) are widely used in electronic devices and components in the field of energy conversions, such as transformers, inductors and electrical motors [1–3]. They are key to the efficient operation of the next generation of electrical machines due to their characteristics, such as magnetic and thermal isotropy, high resistivity and high saturation magnetization [2,4]. In order to be used in high-frequency ranges, SMCs require excellent electromagnetic properties, such as high-frequency stability, low core loss and usability with high currents [5]. The frequency characteristics of permeability and total core loss can be significantly affected by structure, density, non-magnetic insulation coatings, internal stress and so on [6]. Generally, insulation coatings can improve the resistivity of powders and insulate the particles, which can reduce the eddy current loss at high frequency. Meanwhile, the non-magnetic insulation coatings will dilute saturation magnetization and decrease permeability. Furthermore, the internal stress generated during the pressing process will hinder domain walls' motion, resulting in the deterioration of coercivity and hysteresis loss [7–9].

There are many types of SMCs, such as ferrites, FeSi, Sendust (FeSiAl) and Fe-based amorphous cores. Compared with traditional soft magnetic alloys, Fe-based amorphous materials have low coercivity, high resistivity and high saturation magnetization [10]. Fe-based amorphous bulks have been used in transformers and are estimated to be able to save approximately 30% of electrical energy. Therefore, Fe-based amorphous materials have attracted widespread attention and are recommended as ideal soft materials for high-frequency applications. However, due to the poor plastic deformation ability of amorphous powders, they require higher pressure during pressing than compared to traditional SMCs. The internal stress generated during the pressing process leads to low permeability and high core loss. In this research, FeSiBCr amorphous magnetic powder cores (AMPCs) with phosphate–resin hybrid insulation coating were fabricated. The annealing effects on the microstructure and magnetic properties of the powder cores were studied systematically.

## 2. Experimental Procedure

The FeSiBCr gas-atomized amorphous powders with a median particle size of 11 μm were first mixed with phosphoric acid diluted in ethanol with a concentration of 0.6 wt.%. The powders were stirred in the solution at 55 °C for 30 min to obtain the phosphate coating. Then, the coated powders were dried at 120 °C for 30 min. After drying, the coated powders were mixed with 3 wt.% epoxy-modified silicone resin acetone solution to obtain inorganic–organic core–shell composite powders. The composite powders were then compacted into toroidal cores with dimensions of Φ20 mm × Φ12 mm × 5 mm at 1800 MPa. The coated amorphous powder and the powder cores were annealed at 440 °C, 480 °C and 520 °C under the protection of argon for 1 h.

The morphology of the coated powders was characterized by a scanning electron microscope (SEM). The phase identifications for all powders were conducted by an X-ray diffractometer using Cu Kα radiation over the 2θ range of 10°–90°. The saturation magnetizations of all samples were measured with a Physical Property Measurement System (PPMS) equipped with a vibrating sample magnetometer (VSM). The DC hysteresis loops of all powder cores were collected by a soft magnetic direct current test system. The effective permeabilities and quality factors of the powder cores were measured by an impedance analyzer. The core losses of all samples were measured using an AC B-H loop tracer.

## 3. Results and Discussion

Figure 1 shows SEM images of the raw amorphous powder, the phosphated powder and the phosphated powders after annealing at different temperatures. After phosphating, the surfaces of the powders became rougher than that of the raw powders. However, the morphology of the phosphated powders annealed below 480 °C did not reflect significant changes. In particular, it can be seen in Figure 1e that the surface morphology of the powder annealed at 520 °C shows a number of pits due to degradation of phosphate [11].

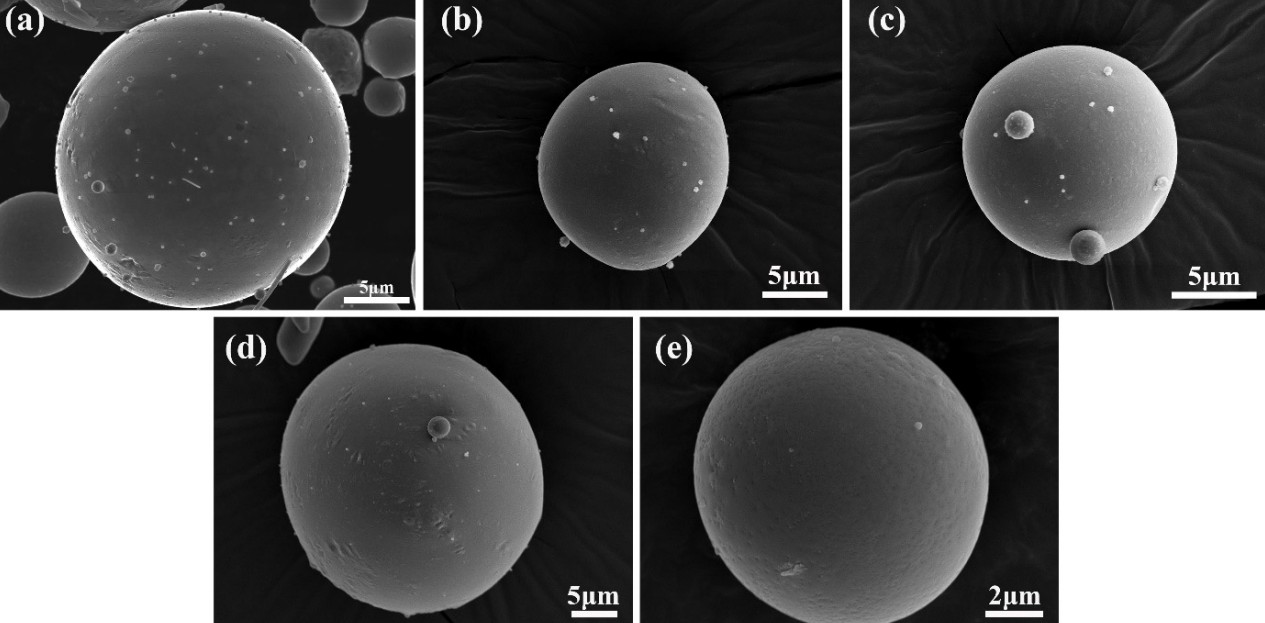

**Figure 1.** (**a**) SEM images of raw amorphous powder, (**b**) phosphated amorphous powder, (**c–e**) the phosphated powder after annealing at 440 °C, 440 °C and 520 °C, respectively.

The thermal stability of the amorphous powder was investigated. Figure 2 shows the DSC curve where only one exothermic peak at $T_x$ = 548 °C was observed, which may

correspond to the crystallization of α-Fe (Si). Details of the crystallization will be discussed in the following section.

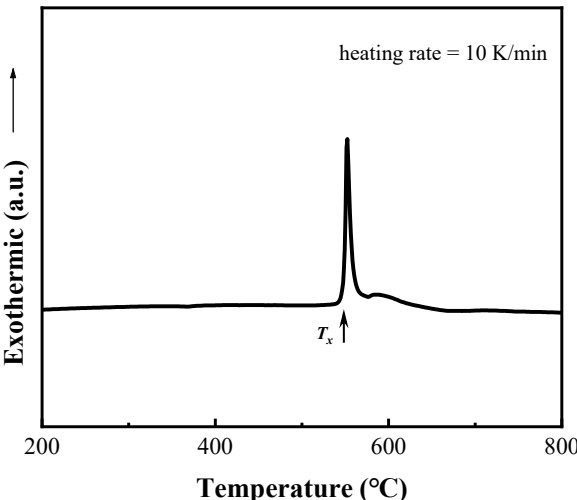

**Figure 2.** DSC curve of the raw FeSiBCr amorphous powder with a heating rate of 10 K/min.

The XRD patterns of phosphated powders after annealing at different temperatures for 1 h are shown in Figure 3. The patterns of the raw powder and phosphate powders are also presented for a comparison shown in Figure 3. Owing to the low concentration of phosphate, the pattern of the phosphated amorphous powders presents the same amorphous peak as the raw amorphous powders. For phosphated powders annealed at temperatures below 480 °C, no crystallization peaks can be observed. However, the crystallization peaks of the α-Fe (Si) phase appeared after annealing at 520 °C, indicating that the α-Fe (Si) phase precipitated from the amorphous matrix. Additional peaks of $Fe_3B$ were also detected for the sample annealed at 520 °C.

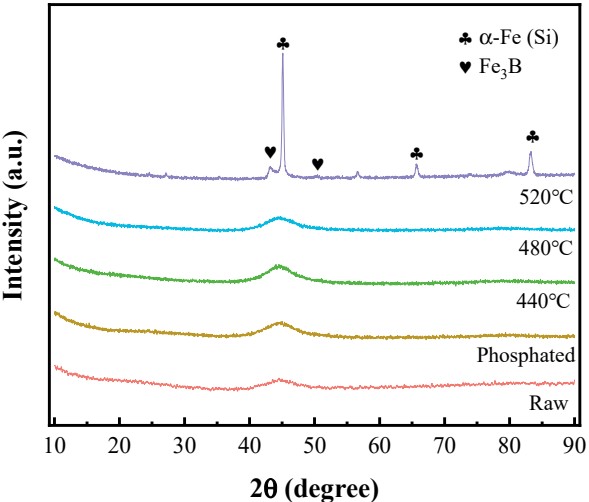

**Figure 3.** The XRD patterns of raw powders, phosphated powders and powders annealed at 440 °C, 480 °C and 520 °C.

Figure 4a shows the magnetization of the raw amorphous powder, phosphated powder and the powders after annealing at different temperatures. Compared with the raw powder, the saturation magnetization of phosphated powders is lower, which is attributed to the magnetic dilution of non-magnetic phosphate on the surface of the amorphous powders after phosphating [12,13]. After annealing at 440 °C, 480 °C and 520 °C, the saturation magnetizations of the composite powders are 147 emu/g, 150 emu/g and 151 emu/g.

This is due to the formation of α-Fe in the amorphous matrix and the degradation of the phosphate coating, which increases saturation magnetization [11,14]. The DC magnetic hysteresis loops of the amorphous powder cores before and after annealing at 440 °C, 480 °C and 520 °C are shown in Figure 4b. After annealing at temperatures below 480 °C, the slope of the DC hysteresis loop increases, and the area of the loop becomes smaller. As can be seen in Figure 4c, the loops in external magnetic fields of $H = -40$ to 40 Oe demonstrate that the coercivity of the unannealed amorphous powder core is 520 A/m. Meanwhile, the coercivities of the powder cores annealed at 440 °C and 480 °C are 39.3 A/m and 39.6 A/m, respectively. This is because the internal stress of the amorphous powder cores generated during cold pressing was released due to the annealing heat treatment [15]. With an increase in annealing temperature to 520 °C, the slope of the DC hysteresis loop decreases, and the loop area becomes wider than the loop of the unannealed powder cores. Combined with the results of XRD in Figure 3, after annealing at 520 °C, α-Fe and $Fe_3B$ phases appear in the amorphous matrix. The grain boundaries generated by these crystals will hinder the movement of domain walls, which will result in a deterioration of coercivity. The corresponding saturation magnetizations and coercivities of the powder cores are shown in Figure 4d.

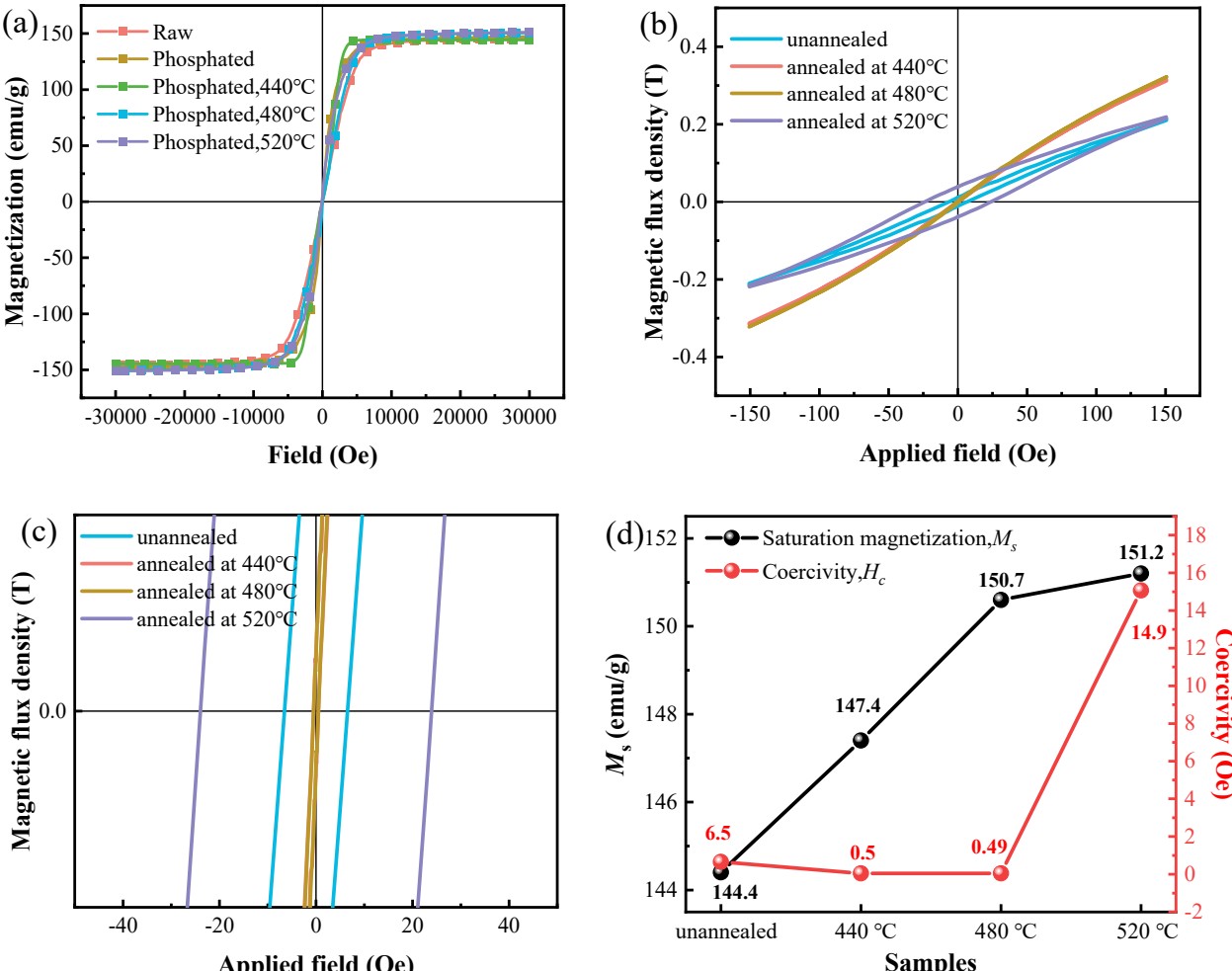

**Figure 4.** (**a**) *M-H* curves of the raw powders, coated powders and powders annealed at different temperatures; (**b,c**) show the DC magnetic hysteresis loop and the partial loops of the AMPCs in field $H = \pm 150$ Oe; (**d**) saturation magnetization $M_s$ and coercivity $H_c$ of AMPCs before and after being annealed at different temperatures.

Within the frequency range from 10 kHz to 110 MHz, the relationship between the effective permeability and frequency is illustrated in Figure 5a for the powder cores before and after annealing at different temperatures. As observed in Figure 5a, for the unannealed amorphous powder core, the effective permeability is 20.4 at low frequencies. With an increase in frequency, effective permeability presents a dispersion phenomenon [16], meaning that the effective permeability remains stable until 17 MHz. After annealing at 440 °C, the effective permeability increases to 34.3 at low frequency. With an increase in annealing temperature to 480 °C, effective permeability slightly increases to 36.4 at low frequencies. At low frequencies, the reversible magnetic domain wall movement and the reversible magnetic moment rotation contribute to effective permeability [2,7,17]. As shown in Figure 4, after annealing at 440 °C and 480 °C, the internal stress within the particles was almost all removed, which is attributed to the markedly increasing effective permeability. Although effective permeability was greatly improved, the frequency stability of effective permeability was greatly reduced. Compared with the unannealed powder core, the frequency of the effective permeability stability for powder cores annealed at 440 °C and 480 °C deteriorated to 3.2 MHz. After further increaseing the annealing temperature to 520 °C, effective permeability decreased dramatically to 20.8 and remained stable until 110 MHz. The grain boundaries within the particles annealed at 520 °C will hinder the movement of magnetic domain walls, such as internal stress, resulting in a decline in effective permeability. Figure 5b shows the frequency dependence of the quality factor *Q* for the powder cores after annealing at different temperatures. Usually, quality factor *Q* represents the efficiency of energy utilization. The *Q* values for the powder cores annealed at 440 °C and 480 °C have peaks of about 45.5 and 44, respectively, at 2 MHz. With an increase in annealing temperature to 520 °C, the peak value of *Q* decreases to 40.3, but the corresponding frequency moves to 8.5 MHz. Compared with the unannealed powder core, the powder cores annealed at 440 °C and 480 °C have larger peak values of *Q*, indicating that they have better energy utilization [18]. In addition, the lower the effective permeability falls, the larger the frequency corresponding to the peak value of *Q*, indicating that the *Q-f* curve shifts to the right. The overall shift of *Q* to the right indicates that the powder cores annealed at 440 °C and 480 °C have better frequency stability [19].

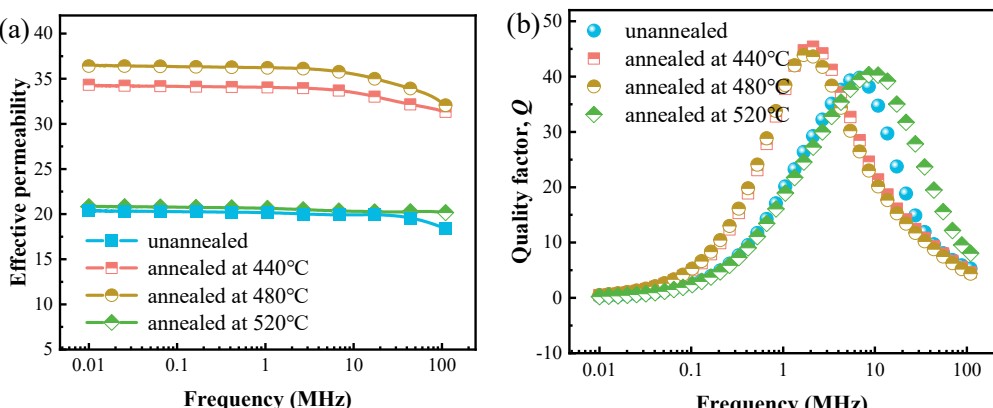

**Figure 5.** The magnetic properties of the AMPCs: (**a**) effective permeability $\mu_e$ versus frequency for amorphous powder cores before and after annealing at different temperatures; (**b**) quality factor *Q*.

Figure 6a shows core loss versus the induction of unannealed powder cores and after annealing at different temperatures at a frequency of 50 kHz. With elevating $B_m$, the core loss of all powder cores increased gradually, and the minimum core loss was obtained for the samples annealed at 440 °C and 480 °C. As the temperature increases to 520 °C, the core loss for the powder cores shows a noticeably higher value because it has greater coercivity due to the crystallization behavior, which hinders the movement of domain walls. Figure 6b shows the core loss in 20 kHz–800 kHz at 10 mT for all samples. With increasing annealing temperature, core losses for all powder cores decreased and then

increased dramatically. The samples annealed at 440 °C and 480 °C exhibited the lowest core losses of about 55.6 mW/cm$^3$ and 51 mW/cm$^3$, respectively, at 800 kHz. In order to analyze the effect of annealing on core loss, loss separation was performed. The total core loss can be presented as follows [16,20,21]:

$$P_t = P_h + P_c = K_h \times f + K_e \times f^2 \tag{1}$$

where $K_h$ and $K_e$ are the coefficients for hysteresis loss $P_h$ and eddy current loss $P_e$, and $f$ is the frequency. The total core loss has been separated and plotted in Figure 6c,d, respectively. Figure 6c shows the variation in hysteresis loss with frequency for all samples. Obviously, the hysteresis loss of all samples first decreased and then increased dramatically with increasing annealing temperature, which has the same trend as the total loss. The sample annealed at 480 °C exhibits the lowest hysteresis loss of about 29.6 mW/cm$^3$ at 800 kHz, and the highest hysteresis loss (approximately 213 mW/cm$^3$ at 800 kHz) was obtained by the sample annealed at 520 °C. Figure 6d shows the frequency dependence of eddy current loss for all powder cores. The eddy current loss for all powder cores increases with frequency in a nonlinear fashion. The unannealed sample has the highest eddy current loss of about 32 mW/cm$^3$ at 800 kHz, while the eddy current loss for the samples annealed at different temperatures is lower. It can be seen from Figure 6c,d that the total core loss of the unannealed sample and the sample annealed at 520 °C is dominated by the hysteresis component (213 mW/cm$^3$ at 800 kHz and 146 mW/cm$^3$ at 800 kHz, respectively) within the measurement frequency range. Owing to hysteresis loss induced by irreversible domain wall movement, internal stress and the grain boundaries generated during high-temperature annealing will hinder the motion of domain walls. Therefore, after annealing, the magnetic performance of the amorphous powder core improved.

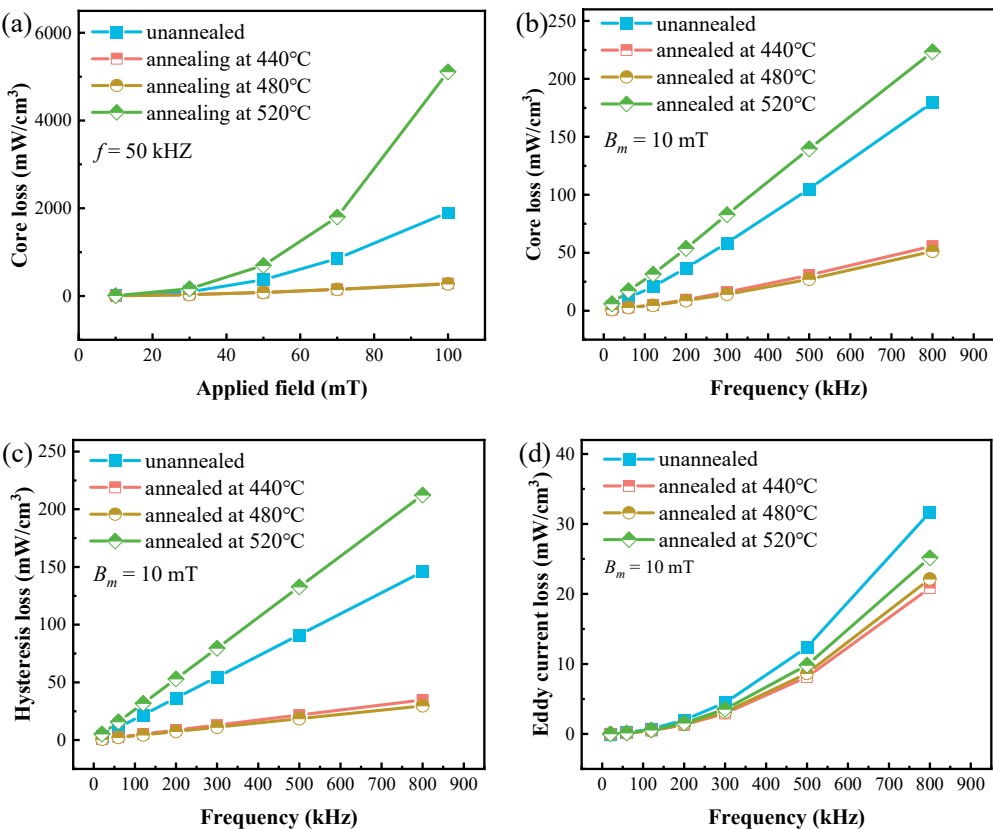

**Figure 6.** (**a**) Core loss versus induction of powder cores after annealing at different temperatures; (**b**) core loss versus frequency of powder cores before and after annealing at 440 °C, 480 °C and 520 °C in 20–800 kHz, at 10 mT; (**c**) hysteresis loss at 10 mT; (**d**) eddy current loss at 10 mT.

Table 1 summarizes the soft magnetic properties of the AMPCs in this paper and the typical SMCs previously reported in the literature [16,18,22–25]. It can be seen that the AMPCs in this paper maintained a relatively lower core loss of 29.6 mW/cm$^3$ at 800 kHz and 10 mT than the other SMCs in Table 1. Furthermore, the effective permeability of the AMPCs is about 36.4 and remains stable until 3 MHz. The improvements in permeability and core loss can expand the usage of AMPCs for high-frequency power applications.

**Table 1.** The core loss of the samples and other research in the literature.

| Sample | $\mu_e$ | $Q_{max}$ | $P_c$ (mW/cm$^3$) | | | References |
|---|---|---|---|---|---|---|
| | | | 0.01 T/800 kHz | 0.05 T/100 kHz | 0.1 T/100 kHz | |
| FeSiBCr@Phosphate | 36.4 | 44 | 51.3 | 152 | 530.4 | This work |
| FeSiBPC@Fe$_3$O$_4$@EP | 49.5 | 160 | - | 187 | 630 | [15] |
| FeSiBCCr@TiO$_2$ | 81.5 | 102 | - | 275 | 900 | [19] |
| FeSiBP@(NiZn)Fe$_2$O$_4$ | 70 | - | - | - | 1000 | [20] |
| FeSiBPNbCr@PPX | 48 | - | - | 220 | 770 | [21] |
| FeSiBP | 86 | - | - | 200 | 780 | [22] |
| FeSiCr@MnZn | 48 | - | 45 | - | - | [13] |

## 4. Conclusions

The magnetic performance of FeSiBCr amorphous powder cores with phosphate–resin hybrid coating was significantly improved by annealing. After annealing at 440 °C and 480 °C, the internal stress within the particles was almost all released. Coercivity decreases markedly, and effective permeability increases significantly and remains stable until 3 MHz. The samples have excellent core loss at 800 kHz due to the significant reduction in hysteresis core loss, while eddy current loss remains very low after annealing below 480 °C. When the temperature increases to 520 °C, the powder cores crystallize, resulting in a deterioration of coercivity, effective permeability and core loss. The low core loss and good frequency stability of the amorphous powder cores provide broad prospects for their application in electronic components at high frequencies.

**Author Contributions:** Conceptualization, H.Y., J.L. (Jiaming Li), J.L. (Jingzhou Li), X.C., G.H., J.Y., R.C.; methodology and data curation, J.L. (Jiaming Li); writing—original draft preparation, J.L. (Jiaming Li); supervision, H.Y. All authors have read and agreed to the published version of the manuscript.

**Funding:** This work was supported by the Dongguan Innovative Research Team Program (Grant No. 2020607231010) and Guangdong Provincial Natural Science Foundation of China (No. 2021A1515010642).

**Conflicts of Interest:** The authors declare no conflict of interest.

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
