# Peer review of "Enhancing the Properties of FeSiBCr Amorphous Soft Magnetic Composites by Annealing Treatments"

_metals, doi:10.3390/met12050828_

Round 1
Reviewer 1 Report
The topic is interesting and refreshing. However, a series of flaws arise that I will address below:
1-The temperature effects are not discussed properly. I think also graphs require more elucidation.
2-Line 141-146 is misleading.
3-I can't really see in the graphs any comparison with most recent works done in literature.
4-Line 183: Therefore, it can be concluded that proper annealing can prove the magnetic performance of the amorphous powder core. What does that mean? if the authors mean "improve", this is a well-known fact.
5-The conclusion part must b more extended and the implications of the present work in applied physics, e.g. electromagnetic applications, semiconductors, etc.
Author Response
1. Thanks for suggestion, i have added more discussion about the effects.
2. The description of the sentence have been changed, it is more easy to read. (see line 141-146 in the re-uploaded manuscript )
3. The graphs is my works, and the table list my works and the resent works.
4. Yes, I mean the magnetic properties improved.
5.Thanks for suggestion, I have extended the conclusion.

Reviewer 2 Report
This paper reported magnetic properties of amorphous powder cores by FeSiBCr with phosphate-resin hybrid coating.
It is interesting in the field of magnetic materials and should be publication.
However, the following points need to be corrected and added.
In page 2 and Figure 3, description of α-Fe(Si) is not explained. Please input explanation in manuscript the α-Fe(Si) meaning.
In DSC, transition temperature shows 548℃ but X-ray diffraction results show crystallization at 520℃. Please explain the differences.
For Figure 4 (c), please input the values at vertical axis. Present figure shows only 0.0.
For Figure 4(d), please confirm the figure. Present figure does not relate to annealing temperature.
Author Response
1. the α-Fe (Si) phase is the Ferrite.
2. the crystallization temperature under 10 K/min heating rate is 548 ℃, but the crystallization behavior of amorphous powder is related with the heating rate of the annealing. The higher the heating rate, the lower of the crystallization temperature. In this work , the amorphous powder crystallized under the heating rate.
3. Fig. 4(c) and (d) have been corrected.

Reviewer 3 Report
In this work, the authors used FeSiBCr amorphous powders with the phosphate-resin hybrid coating to produce Fe-based amorphous powder cores (AMPCs). The study presented in this research is sound, and the results produced are interesting. The paper is sufficiently novel to meet the requirements of the Metals journal. But a major revision is required, and after responding to the following remarks and revising the paper, the manuscript may be considered for publication in the Metals journal.
- The Abstract mainly contains an enumeration of methods, but there is no general information on the results achieved. The abstract should summarize the findings of the work.
- The novelty of the work is missing in the introduction. Authors are suggested to include a separate paragraph by discussing the novelty and importance of the present work.
- The recent literature review is missing in the introduction section. Authors are suggesting to re-write the introduction section by reviewing recent relevant literature in a separate paragraph.
- Authors are suggested to use more references from recent past, and recommend to cite following all references in the introduction section: 10.1016/j.apt.2021.11.030; 10.4236/msa.2015.612108; 10.1016/j.jmmm.2022.169365; 10.4236/wjnse.2015.54013.
- Several experimental instruments were used in this work. It is suggested to include all instruments details, like the model, and origin of the instruments in the Experimental sections.
- The quality of Figures 2-6 is low, please increase the quality. If necessary, redraw them.
- The font size inside all figures is not clearly visible, increase the font size to make it visible.
- Check the typoes throughout the manuscript during revision submission.
Author Response
Revised item by item based on review comments

Round 2
Reviewer 1 Report
I recommend tje publication of the present manuscript.
Reviewer 3 Report
Thanks for the revision.